# *Mirage2*'s high-quality spliced protein-to-genome mappings produce accurate multiple-sequence alignments of isoforms

Alexander J. Nord[1]*, Travis J. Wheeler[2,3]

1 Division of Biological Sciences, University of Montana, Missoula, Montana, United States of America, 2 Department of Pharmacology & Toxicology, University of Arizona, Tucson, Arizona, United States of America, 3 Department of Computer Science, University of Montana, Missoula, Montana, United States of America

* alexander.nord@umontana.edu

**Data Availability Statement:** All data and scripts required to reproduce analyses can be found at: https://osf.io/7pgav/.

**Funding:** A.J.N. and T.J.W acknowledge funding from the National Institute of General Medical

## Abstract

The organization of homologous protein sequences into multiple sequence alignments (MSAs) is a cornerstone of modern analysis of proteins. Recent focus on the importance of alternatively-spliced isoforms in disease and cell biology has highlighted the need for MSA software that can appropriately account for isoforms and the exon-length insertions or deletions that isoforms may have relative to each other. We previously developed *Mirage*, a software package for generating MSAs for isoforms spanning multiple species. Here, we present *Mirage2*, which retains the fundamental algorithms of the original *Mirage* implementation while providing substantially improved translated mapping and improving several aspects of usability. We demonstrate that *Mirage2* is highly effective at mapping proteins to their encoding exons, and that these protein-genome mappings lead to extremely accurate intron-aware alignments. Additionally, *Mirage2* implements a number of engineering improvements that simplify installation and use.

## Introduction

Annotating the structures and functions of protein-coding genes has been a central project of modern biology for as long as researchers have been able to produce nucleotide and amino acid sequences. While the community has achieved rough consensus that there are approximately 20,000 families of protein-coding genes in the human genome [1], this number does not reflect the myriad sequence variations observed within families due to alternative splicing events, through which the same pre-mRNA transcripts can be processed into distinct mRNA sequences that thereby encode unique amino acid sequences (alternative gene isoforms) [2]. Almost all protein families (at least 92% [3]) can be expressed as two or more isoforms due to variations in pre-mRNA processing, with 7 isoforms as the observed average across human genes [4].

By encoding multiple gene isoforms, eukaryotic organisms are able to optimize the particular functions of their genes on the basis of factors like cell specialization and environmental

Sciences (NIGMS, https://www.nigms.nih.gov/), National Institutes of Health (NIH), grants GM123487 and GM132600, and from the National Human Genome Research Institute (NHGRI, NIH, https://www.genome.gov/), grant HG012283. The funders had no role in study design, data collection and analysis, decision to publish, or preparation of the manuscript.

**Competing interests:** The authors have declared that no competing interests exist.

stimulation, because the sequence variations between isoforms translate into diversity in protein structure and function [5, 6]. For biologists studying the functional impacts of alternative splicing, identifying precise differences between isoforms' amino acid sequences can be an essential step towards understanding their divergent behaviors, and one of the fundamental methods for exploring relationships between isoforms (both within and across species) is through the use of multiple sequence alignments (MSAs).

In an MSA, related sequences are aligned in a matrix such that each row represents a single input sequence and each column represents a set of homologous (or identical) residues (nucleotides or amino acid letters) [7]. For isoforms belonging to a single organism, an optimal isoform MSA should accurately reflect the exonic structures of the input sequences, such that amino acids translated from the same coding regions on the genome are aligned, while exon-length gaps indicate where alternative splicing events created variations between the sequences. MSAs for isoforms from multiple species should retain this trait for intra-species aspects of the MSA, and generally obey the same constraints across species, allowing for mutation. One context in which we have used MSAs of isoforms is to aid in the identification of potential targets of post-translational modification in multiple isoforms across multiple species [8].

We previously developed *Mirage* [9], a software package for generating MSAs for isoforms spanning multiple organisms by first mapping each isoform sequence to its coding nucleotides on the genome and then using those nucleotide mapping coordinates as the basis for sequence alignment (Fig 1). Here, we present *Mirage2*, which retains the fundamental algorithms of the original *Mirage* implementation while benefiting from a substantial overhaul of several core components, resulting in a software package that substantially improves the results of translated (protein-to-genome) mapping, records informative intermediate outputs useful for downstream analysis, and improves several aspects of usability.

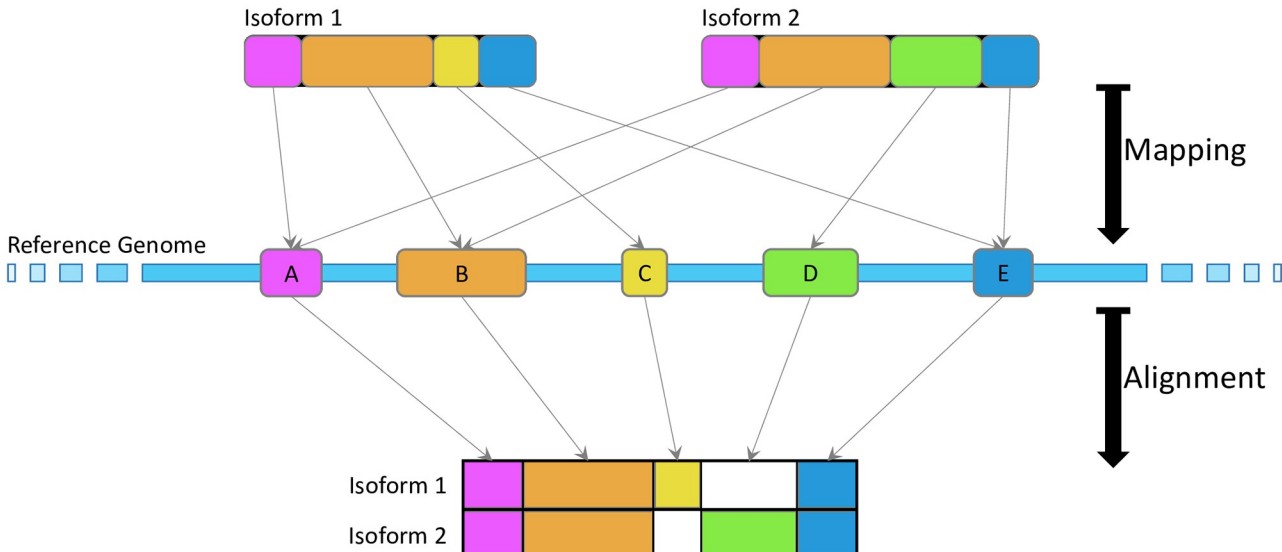

**Fig 1. A schematic of the core *Mirage2* algorithm.** Isoforms are first mapped back to their coding exons on the genome. Once all isoforms within a gene family have been mapped, those genome mapping coordinates serve as the basis for intra-species alignment, resulting in an MSA with explicit splice site awareness and exon delineation.

## Materials and methods

### Inputs: An isoform database and a "species guide"

The primary input to *Mirage2* is a FASTA-formatted database of isoform sequences, and the primary output of *Mirage2* is a collection of MSAs (one MSA per gene family present in the database). To ensure that *Mirage2* correctly groups together sequences belonging to the same gene family, *Mirage2* requires that sequences follow one of two naming conventions that allow the program to easily identify the gene family and species of each sequence. First, sequences can be named using standard *UniProt* [10] conventions, where the GN and OS fields communicate the gene family and species (*e.g.*, >sp|Q5VST9|OBSCN_HUMAN Obscurin OS = Homo sapiens GN = OBSCN). This allows users to simply download sequences directly from *UniProt* for incorporation into their alignments. The second naming convention uses '|' symbols to delineate the sequence's species, gene family, and a unique identifier within that gene family (*e.g.*, human|obscurin|isoform1). Each of these conventions enables *Mirage2* to easily group sequences according to their gene families and identify the appropriate genome to use as a sequence's mapping target.

In addition to the isoform database, *Mirage2* requires the user provide a "species guide" file. The main role of the species guide is to associate each species represented in the isoform database with the location of the FASTA-formatted genome file for that species on the user's system, since clear correspondence between species' names and genomes is essential for *Mirage2* to generate the protein-to-genome mappings on which its alignments are based (Fig 2).

Two optional pieces of additional information that the user can provide in the species guide are a Newick-formatted species tree and GTF-formatted index file locations. The species tree provides an explicit merge order for *Mirage2* when combining intra-species MSAs to produce each gene family's final inter-species MSA. GTF indices are optional species-specific files that provide genomic coordinates for known/predicted coding regions of particular gene families, and whose availability enables *Mirage2* to use its native mapping method, *FastMap*.

### Protein-to-genome mapping using *BLAT* and *Spaln2*

The first stage of *Mirage2*'s pipeline involves mapping each isoform sequence to the set of genomic coding regions (exons) corresponding to the processed mRNA transcript from which the protein was derived. Ideally, *Mirage2* can make use of GTF index files to guide the mapping process (see next section), but we first describe the baseline case in which no GTF index files are available. The translated alignment tool *Spaln2* [11] is an accurate tool for generating heuristic-based spliced alignments of protein sequences to nucleotide targets, and is thus a key resource for *Mirage2*'s mapping stage. However, *Spaln2* is too slow to use for aligning each

```
(human,(mouse,rat))
human   /data/genomes/GRCh38.fa     /data/gtfs/human.gtf
mouse   /data/genomes/GRCm38.fa     /data/gtfs/mouse.gtf
rat     /data/genomes/RGSC_6.0.fa   /data/gtfs/rat.gtf
```

**Fig 2. An example of a "species guide" file.** The top line is a Newick-formatted species tree to set the merge order of species during *Mirage2*'s interspecies alignment phase. Each subsequent line associates a species with the location of its reference genome and a GTF index. Sequences belonging to species that aren't listed in the species guide are treated as "miscellaneous" and are the last to be integrated into interspecies MSAs.

sequence to an entire genome (for example, it takes just over 5 minutes to search a single 200 amino acid sequence against the human genome), so *Mirage2* first uses *BLAT* [12] to quickly search each protein against its genome.

*BLAT* is a translated alignment tool that is capable of rapidly computing partial alignments between protein sequences and a large nucleotide target (*e.g.*, genome), albeit at the computational cost of generating an index on the nucleotide target during initialization. The collection of partial-length alignments produced by *BLAT* can be thought of as protein-specific exon predictions that enable *Mirage2* to identify a relatively narrow window of genomic sequence as the putative coding region for each protein sequence provided to *BLAT*. *Mirage2* then takes each sequence's *BLAT*-predicted putative coding region and widens it by 50,000 nucleotides in both directions. This establishes a robust search region that is still sufficiently constrained to be provided as an input to *Spaln2*, thus enabling *Spaln2* to quickly compute a full protein-length spliced alignment between the isoform and its putative coding region. Finally, *Mirage2* performs modest post-processing on the spliced alignments produced by *Spaln2* to clarify splice boundaries and nucleotide coordinates (see Discussion) and records the processed alignments as sets of mapping coordinates, wherein each amino acid in an isoform sequence is assigned to the genomic coordinate of its codon's center nucleotide.

## GTF indices enable enhanced mapping methods

While *BLAT* is a fast tool for whole-genome search, searching each sequence in a large isoform database against a genome can still prove computationally expensive. In order to reduce the number of sequences searched against the entire genome using *BLAT*, *Mirage2* allows users to associate each genome with a GTF index file, which will trigger a pre-*BLAT* search phase. GTF indices list various types of genomic annotations, including the nucleotide coordinates of exons known (or predicted) to be used in particular gene families' proteins. A gene family's GTF-annotated exon coordinates can thus serve the same fundamental purpose as a set of *BLAT* search results—they highlight the window of genomic sequence wherein we expect to find optimal mappings for a family's isoform sequences. *Mirage2* takes advantage of the information in GTF files in two ways, first performing exon-guided mapping with its native tool *FastMap*, then (if that mapping attempt fails) mapping to the GTF-guided region with *Spaln2*.

Here, we briefly describe the *FastMap* tool in additional detail. The input to *FastMap* is a single protein sequence $p$ from species $q$ and a set $T$ of DNA ranges that make up the complete set of exons listed in $Q$'s gene index file for protein $P$. For each sequence $t$ corresponding to ranges in $T$, *FastMap* aligns $p$ to $t$ with a dynamic programming algorithm that is local with respect to $p$ and global with respect to $t$. Rather than filling the entire dynamic programming matrix, only near-perfect ungapped alignments are stored and computed, so that space and run time are generally linear in the length of $t$. Note that it is common for GTF files to contain many overlapping proposed exons. Mapped exons are stitched together with a native tool called *ExonWeaver*, which traverses a splice-graph inferred from those (potentially overlapping) sub-sequence alignments to produce an optimal mapping of the full-length isoform to the genome.

Not only is this native pipeline extremely fast, but it avoids a small handful of bugs that we have observed in *Spaln2* which require a modest amount of post-processing to adjust for. Moreover, *Mirage2*'s native spliced alignment pipeline enjoys the epistemic benefit of using known coding regions as its alignment targets, rather than computationally-predicted open reading frames. If *FastMap* fails to completely map the sequence $p$ to the genome, then *Mirage2* falls back to running *Spaln2* on a window extracted around the GTF range. If this fails (or if no GTF file was provided), *Mirage2* uses the *BLAT*+*Spaln2* strategy described above.

*Mirage2* optionally produces the results of this mapping stage; we have found this output useful, for example, in exploring uncommon splicing activity.

### Generating intra-species MSAs *via* mapping coordinates

Following the mapping stage, isoform sequences that fully mapped back to their genomes and belong to the same species and gene family are organized into an MSA using their genome mapping coordinates. To achieve this, a hash is created where the hash keys are nucleotide coordinates and the hash values are list of tuples, with each tuple consisting of a numerical sequence identifier (*i.e.* the MSA row corresponding to the isoform sequence) and the amino acid that was mapped to the key nucleotide coordinate from that sequence. Once all sequences have been written into the hash, each entry in the hash table represents a single MSA column, such that an in-order traversal of the sorted hash keys allows for straightforward MSA construction. Additionally, during the traversal of the hash keys that generates the MSA, whenever the absolute difference between two hash keys is greater than 10 (nucleotides) a column of '/' (forward slash) characters is placed in the MSA to indicate the location of an inferred intron (a "splice column")—these slash columns are used for inter-species alignment, described in the next section. After all mapped sequences are aligned, sequences that failed to map back to the genome are individually incorporated into their intra-species MSAs using an implementation of Needleman-Wunsch global alignment [13].

### Merging splice-aware MSAs across species

Following intra-species MSA generation, MSAs belonging to the same gene family are merged across species using a straightforward profile-to-profile alignment [14] algorithm. Our method prohibits alignment of splice-columns to amino acid characters, and also employs scoring heuristics that restrict when the sequences attributed to a single exon in one species might be aligned to multiple sequences of exons from another species (suggesting either intron deletion in the first species, intron creation in the second species, or shift in splice site location). Specifically, match scores between residues are computed using half-bit BLOSUM62 [15] scores, gaps are scored with an affine gap penalty (gap open = −11, gap extension = −1), and a gap at a splice site incurs a penalty that scales with the distance $d$ to the closest splice site in the other MSA under the function $MAX(-5 \cdot 2^d, -200)$.

If a species tree was provided in the species guide file, intra-species MSAs will be merged in an order that traces from the leaves of the tree to its root. Species that are not listed in a species tree are merged into the inter-species MSA in no particular order. The result of merging the intra-species MSAs is that each gene family has a single inter-species MSA constructed to optimally represent exon-to-exon alignment.

## Results

### Testing setup

We performed all testing on the complete set of human, mouse, and rat protein sequences available in the *UniProt*KB/*SwissProt* dataset [10]. *SwissProt* is a curated set of protein sequences that are trusted to be expressed as genuine *in vivo* proteins, and from this collection we derived a set of 73,573 human, mouse, and rat protein sequences representing 17,912 gene families.

We used the GRCh38, GRCm38, and RGSC 6.0 reference genomes available through the UCSC Genome Browser [16] as the targets for protein-to-genome mapping. We constructed GTF indices by concatenating the RefSeq and RefGene indices produced by Ensembl [17]

available for each species on the UCSC Genome Browser. Although this may create redundancy in our set of known coding regions, it ensures that *Mirage2* has access to the most robust indexing information possible.

## *Mirage2* maps nearly all *SwissProt* sequences to their encoding exons

The core of the *Mirage2* algorithm is the protein-to-genome mapping phase, which enables *Mirage2* to accurately capture exon-level relationships across isoform sequences. For this reason, it is critical to the success of *Mirage2* that it can identify high-quality full-length protein-to-genome mappings for as many sequences as possible. *Mirage2* successfully maps 99.12% of the sequences in our *SwissProt* dataset back to their full set of coding exons on the genome (Table 1). The vast majority of these mappings are produced by *Mirage2*'s native mapping method, *FastMap*, and only 2.9% of sequences required *Mirage2* to perform a computationally expensive *BLAT* search in order to identify a complete coding region on the genome.

    *Mirage2* performs sequence mapping via a cascade of steps, first applying a fast exon-based mapping strategy guided by all potential exons found in the input GTF file (*FastMap*), then running *Spaln2* guided by GTF-informed coordinates on unmapped sequences, and finally mapping remaining sequences by seeking a mapping seed with *BLAT* and applying *Spaln2* in a window around that seed (see Methods for details). In order to maximize the speed and accuracy of its mapping phase, *Mirage2* gives preference to its native mapping method, *FastMap*.

    To the best of our knowledge, the only other tool for splice-aware mapping of protein sequences to their encoding genome is *Spaln2*, which is incorporated as a subroutine of *Mirage2*. To investigate mapping efficacy of *Spaln2* alone (without *Mirage2*'s *FastMap* tool), we performed a duplicate run of *Mirage2* on the *SwissProt* dataset with *FastMap* disabled. We observed that the majority of sequences could be successfully mapped using either *FastMap* or *Spaln2*, but that in each species roughly 12–15% of mappable sequences required a specific tool (Fig 3). In addition to the speed gains achieved through use of *FastMap*, this highlights the value of basing mapping alignments on exon candidate from potentially noisy GTF files.

## *Mirage2* intra-Species MSAs are extremely high-quality

Establishing reliable and generalizable metrics and datasets for quantitatively comparing MSA methods is notoriously difficult [18, 19], as it is easy for assumptions that implicitly favor one approach over another to filter into proposed means of evaluation. *Mirage2*, however, is designed for the specific application of aligning sequences that are isoforms derived from a common gene, which allows us to ground our comparisons in expectations particular to the nature of isoform production.

    First, we note that within species we can typically expect 100% column identity in an accurate isoform MSA (where an alignment's percent column identity is the percentage of MSA

**Table 1. *Mirage2*'s mapping methods map nearly all *SwissProt* sequences.**

| Mapping Method | Human | Mouse | Rat |
|---|---|---|---|
| *FastMap* | 36,169 | 21,903 | 8,727 |
| *Spaln2* (GTF coord.s) | 2,271 | 1,222 | 354 |
| *Spaln2* (*BLAT* region) | 791 | 1,201 | 344 |
| Unmapped | 284 | 206 | 101 |
| **Mapping Success Rate** | **99.28%** | **99.16%** | **98.94%** |

*Mirage2* maps almost every isoform sequence in the *SwissProt* dataset back to a set of coding exons on the genome.

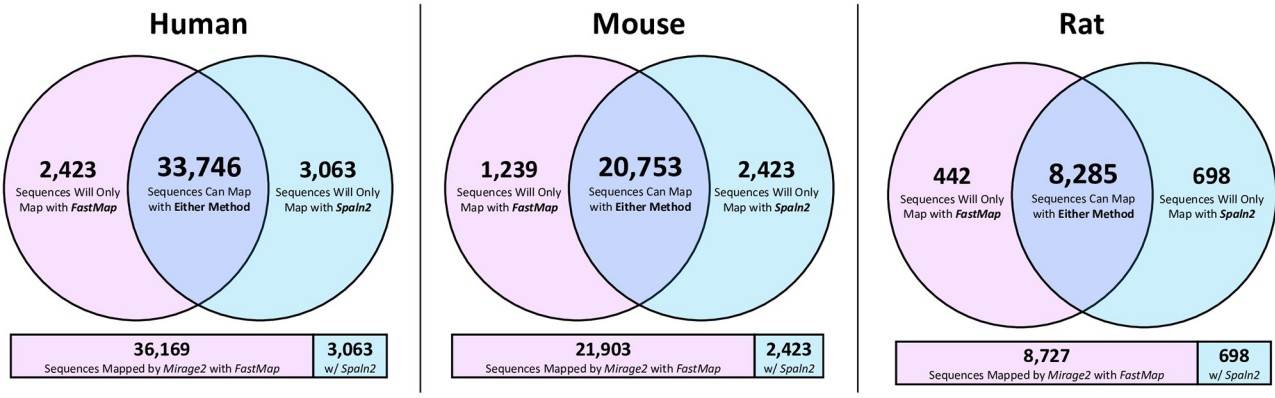

**Fig 3. *FastMap* and *Spaln2* are complementary mapping methods.** The majority of sequences that Mirage2 is able to map back to the genome can be mapped using either *FastMap* or *Spaln2*, although one tool or the other is specifically required to map 14.0% of human sequences, 15.0% of mouse sequences, and 12.1% of rat sequences.

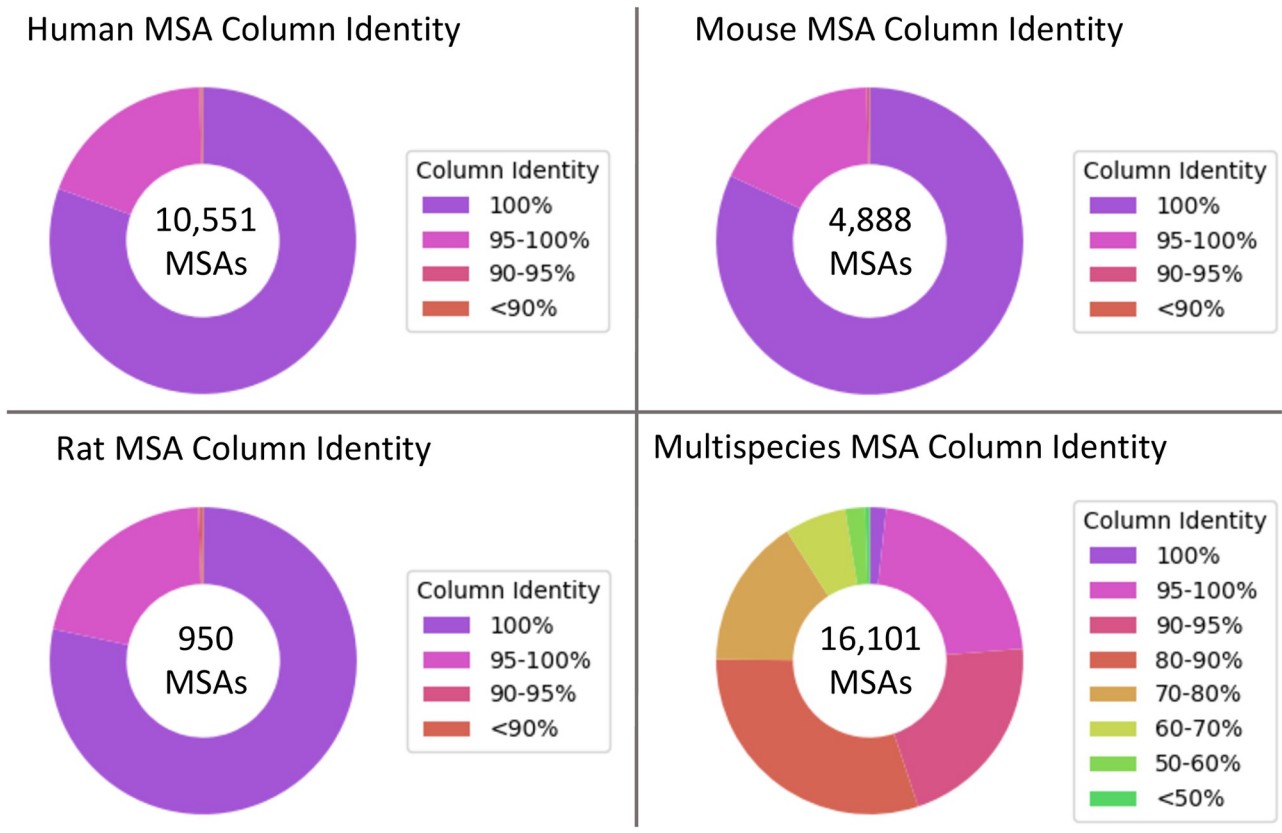

**Fig 4. *Mirage2* MSAs have extremely high percents column identity.** Percent column identity distributions for intra-species *Mirage2* multiple-sequence alignments (excluding "alignments" with only 1 sequence) and *Mirage2* inter-species alignments for genes present in at least 2 species.

columns where all cells contain the same amino acid character, excluding gap characters). The basis of our expectation of 100% column identity is that the sequences in an isoform MSA are translations of exactly the same genomic sequence, varying only in the specific set of exons included in the protein product (ignoring the sort of infrequent post-transcriptional phenomena that may cause minor variations). For each of our three species, we observed that the vast majority of *Mirage2* intra-species MSAs exhibited 100% column identity, and that vanishingly few had column identity lower than 90% (Fig 4).

Comparing the percent column identity of each *Mirage2*'s intra-species MSA to the corresponding alignments generated by the general-purpose MSA tools *MAFFT* [20], *Muscle* [21], and *Clustal-Omega* [22] (Fig 5), we observed that, while most intra-species MSAs show high column identity, a few alignments are much worse than that prduced by *Mirage2* (*e.g.* in human we observed that 11.0% of *Clustal-Omega* alignments show a reduction in percent column identity of at least 10%, with 7.7% of *MAFFT* and 9.9% of *Muscle* alignments also showing a $\geq$10% reduction in column identity).

A similar pattern is observed in our comparison of inter-species MSAs across tools, although we note that raw column percent identity is not as strong of a metric when evaluating inter-species isoform alignment quality, since an accurate exon-aware alignment may exhibit lower column identity in regions where aligning evolutionarily diverged homologous exons requires aligning non-identical residues.

We note that a very small number of alignments generated by general-purpose tools exhibit greater percent column identity than the corresponding *Mirage2* MSA. These are almost entirely cases where none of the sequences in a gene family successfully mapped to the genome, and as such are aligned using a more general-purpose dynamic programming method which, because its primary utility in *Mirage2* is merging exon-aware alignments across species, is generally willing to align chunks of dissimilar sequence so long as there are a handful of identical residues. This approach produces highly accurate MSAs when aligning evolutionarily diverged exons, but is prone to over-aligning dissimilar sequences when it does not have access to information about splice site locations.

In addition to the special expectation of 100% column identity, another unique feature available for comparing MSA tools in the context of isoform alignments is the lengths of the alignments that they produce. The underlying heuristics of general-purpose alignment methods can severely penalize the extended runs of gap characters often necessary for accurately delineating exons. This causes residues in unrelated exons to be incorrectly forced into the same column to optimize the total score of the alignment under the tool's particular scoring scheme. To gain insight into the scale of this problem, we explored the frequency and magnitude of such over-alignment. Let $m$ be the length (number of columns) of a *Mirage2* MSA, and $a$ be the length of the corresponding MSA produced by an alternative tool; we define the "length compaction factor" to be $a/m$. The compaction factor is computed for each gene family, and communicates the extent to which ignoring exons allows alignments to be compacted (Fig 6).

Similar to what we see in our evaluation of percent column identity, we note that there is a very small number of cases where an MSA's extension factor suggests that *Mirage2* is underperforming an alternative tool. As far as we have observed, however, these are rare instances where the alternative tool significantly misaligns certain sequences, resulting in an erroneous alignment whose length is much greater than the corresponding *Mirage2* MSA (Fig 7). A partial survey of these cases suggests a high degree of repetition in the input sequences as a suspicious common factor, but we cannot say for certain why these errors occur for other MSA tools.

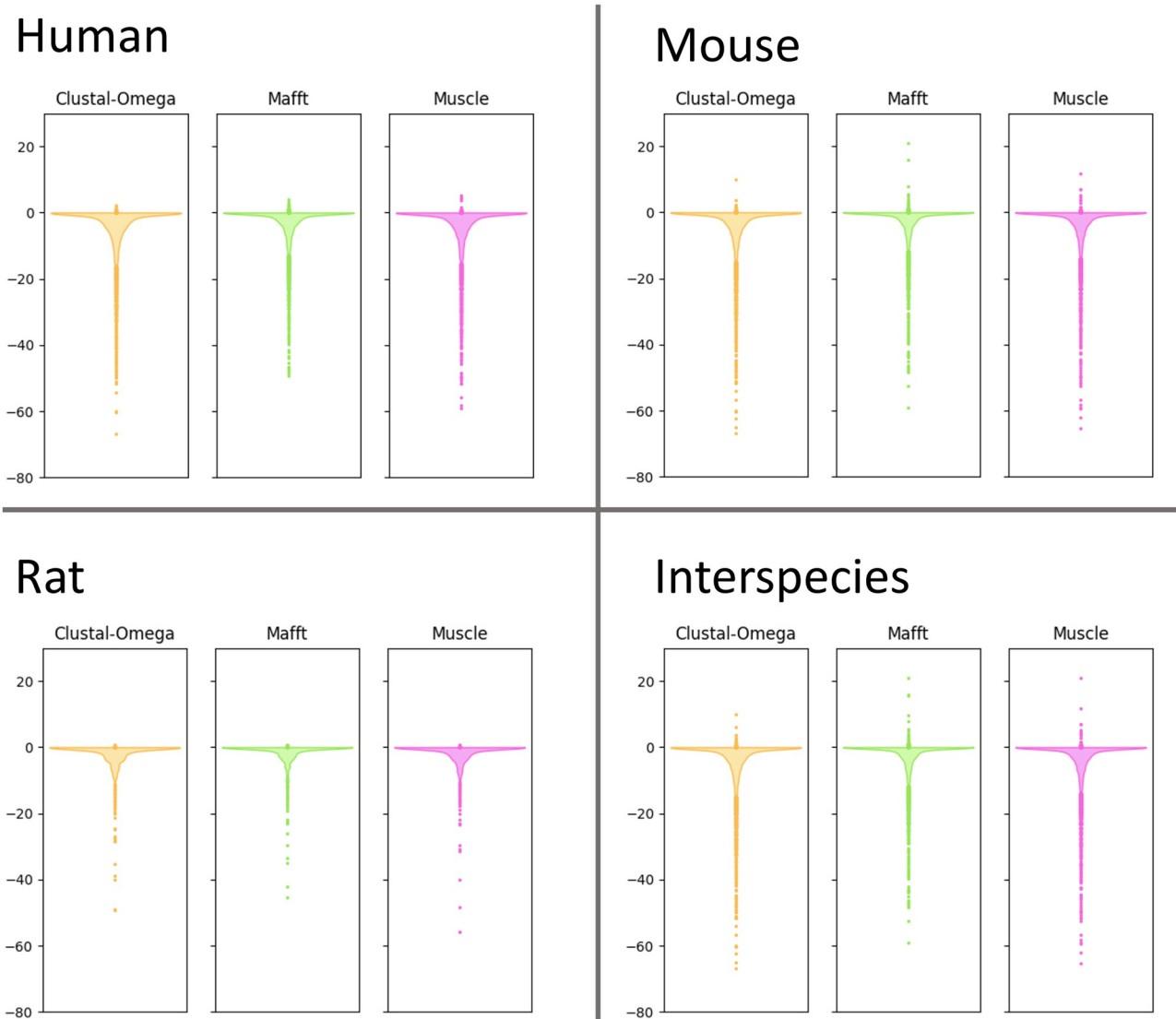

**Fig 5. Differences between the percents column identity of *Mirage2* MSAs and alignments produced by general-purpose MSA tools.** Values were computed by subtracting the percent column identity of each *Mirage2* MSA from the percent column identity of corresponding MSA produced by an alternative tool.

### *Mirage2* improves over the performance of *Mirage*

In addition to improvements to the user experience and the robustness of intermediate program outputs, *Mirage2* exhibits several important performance improvements over the original *Mirage* implementation. The most important of these improvements is an increase in the number of isoform sequences mapped back to the genome for each species, which more than halves the number of unmapped sequences (Table 2). Moreover, there is a significant shift in each species towards the use of *Mirage2*'s native mapping tool, *FastMap*, over *Spaln2* for sequences whose coding region is highlighted by GTF coordinates. Because mappings produced by *FastMap* are informed by known coding regions, this greatly bolsters our confidence in the veracity of the reported mappings.

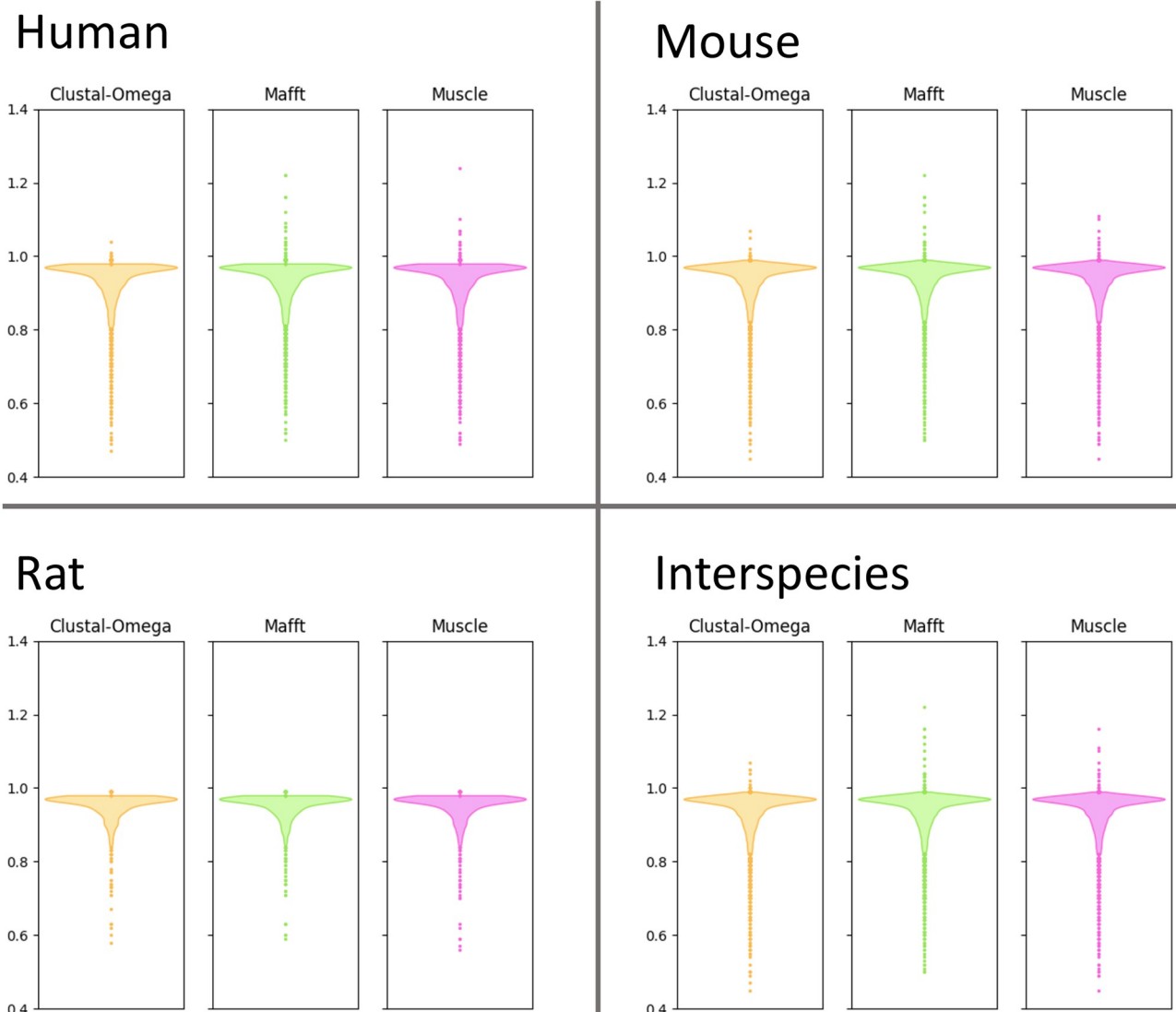

**Fig 6. The length compaction factors of alternative alignment methods relative to *Mirage2*.** Alignment length is defined as the number of columns in an MSA and the compaction factor is computed by dividing the length of an alternative tool's MSA by the length of the corresponding *Mirage2* MSA.

### *Mirage2*'s runtime is comparable to other tools

We ran each of the four tools on the *SwissProt* dataset using default settings and access to 16 Intel® Xeon® E5–2630 compute cores, acquiring the elapsed wall-clock time required for each method to generate inter-species MSAs for all gene families (Table 3).

While *Mirage2*'s runtime was greater than that of any other tool at default settings, the majority of its runtime is attributable to performing whole-genome *BLAT* searches during its mapping phase. *BLAT* search can be disabled using the `--blat_off` commandline flag, which dropped *Mirage2*'s total runtime to 50.5 minutes, making it nearly 50% faster than its closest competitor. The acceleration achieved by deactivating *BLAT*-based search leads to a decrease in the percentage of mapped sequences, from 99.20% to 93.55% of all sequences mapping back to their full coding regions. Examining all 32,511 MSAs with more than one sequence (intra-species and multi-species), 3,634 alignments suffer some decrease in percent

## Mirage2

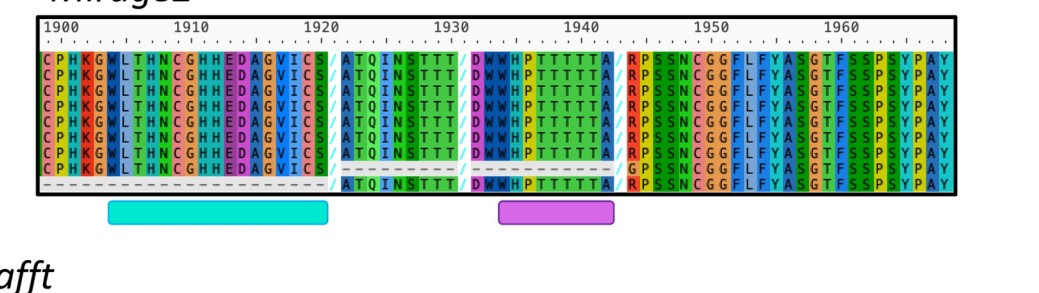

## Mafft

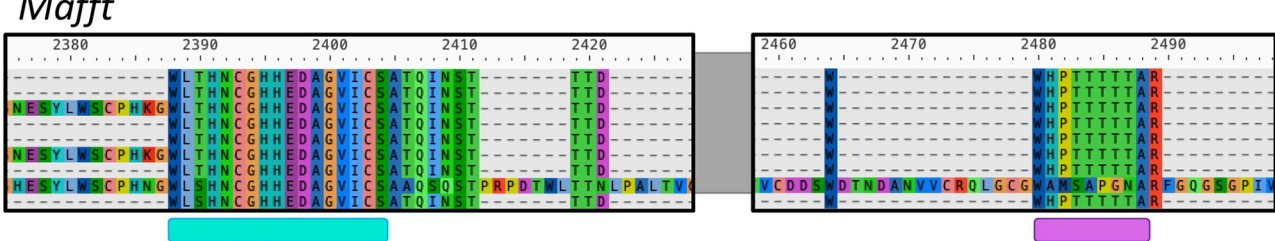

**Fig 7. A partial comparison of the alignments of human DMBT1 sequences produced by *Mirage2* and *MAFFT*.** The underlined segments highlight sequence regions where the tools are generally in agreement, but the segments are spaced significantly further apart in the *MAFFT* alignment than they are in the *Mirage2* alignment. This illustrates how, in cases of erroneous alignment, using the comparative lengths of isoform MSAs can be an imperfect quantification of relative alignment quality.

**Table 2. Mapping performance comparison between *Mirage2* and the original *Mirage* implementation.**

| Mapping Method | Human | | Mouse | | Rat | |
|---|---|---|---|---|---|---|
| | *Mirage2* | *Mirage* | *Mirage2* | *Mirage* | *Mirage2* | *Mirage* |
| *FastMap* | 36,169 | 31,518 | 21,903 | 18,629 | 8,727 | 7,202 |
| *Spaln2* (GTF coord.s) | 2,271 | 7,195 | 1,222 | 4,653 | 354 | 1,933 |
| *Spaln2* (BLAT region) | 791 | 213 | 1,201 | 869 | 344 | 214 |
| Percent Unmapped | **0.72** | 1.68 | **0.84** | 1.57 | **1.06** | 1.87 |

A comparison between the mapping success rates of *Mirage2* and *Mirage* shows an increase in the number of sequences mapped by *Fastmap*, and an overall increase in the percentage of sequences mapped from 98.33% to 99.20%.

**Table 3. Runtime comparison between *Mirage2* and alternative MSA tools.**

| Tool | | Runtime (minutes) |
|---|---|---|
| *Mirage2* | default | 265.0 |
| | `--blat_off` | 50.5 |
| *Clustal-Omega* | | 213.3 |
| *MAFFT* | | 95.3 |
| *Muscle* | | 136.7 |

Total elapsed time required for each tool to generate inter-species MSAs for each of the 17,912 gene families in the *SwissProt* dataset running on 16 compute cores.

column identity, but these decreases are generally negligible, with an average decrease of 2.11% column identity and a median decrease of 0.5% column identity.

## Discussion

### Improved usability

The substantial reworking of the *Mirage* codebase that produced *Mirage2* includes a number of user-focused quality-of-life improvements. First, a supplemental script is included that automates the processes of downloading genomes and GTF indices for user-specified species from the UCSC Genome Browser and generating the "species guide" file used by *Mirage2*. Second, an option has been added that allows *Mirage2* to terminate after computing protein-to-genome mappings, rather than producing MSAs, along with an improved directory system for accessing information such as mapping coordinates and intra-species MSAs. Similarly, the addition of a flag to disable *BLAT* search affords the user a bit of flexibility in terms of prioritizing speed over exhaustiveness. Finally, a number of engineering improvements have been implemented to make setup more intuitive, including the creation of a Docker image as an avenue for consistent access to a stable *Mirage2* build.

### *Spaln2* output requires significant post-processing

To our knowledge, *Spaln2* is the only tool available for producing full-protein-length spliced alignments between an unannotated nucleotide sequence and an isoform sequence. Generally, *Spaln2*'s alignments are high-quality, but we have uncovered a number of nontrivial errors that we developed *Mirage2* to anticipate, test for, and recover from. The most common of these are simple coordinate indexing errors, but we have observed rarer errors such as *Spaln2* spontaneously introducing insertions or deletions that require substantial re-alignment to the proposed coding region, incorrectly calculating the percent alignment identity (*i.e.* the percentage of amino acids in the isoform sequence that are aligned to codons that translate into the same amino acid), and proposing alignments with series of consecutive micro-exons (exons shorter than 5 amino acids) where a secondary search reveals an alternative alignment wherein a single simple coding region provides a straightforward mapping for the amino acids implicated in the micro-exonic region. All of these errors are relatively simple to recognize and recover from, but require a somewhat baroque approach to parsing *Spaln2*'s output. For this reason, we propose that those interested in *Spaln2* consider running it via *Mirage2* to benefit from the error detection and correction provided by *Mirage2*. Specifically, the `--map_only` flag directs *Mirage2* to provide the protein-to-genome mappings that it would otherwise base its alignments on as the primary output of the program.

## Conclusion

*Mirage2* is a multiple-sequence isoform alignment tool that uses protein-to-genome mappings to produce extremely accurate intra-species MSAs, as well as a splice-aware profile-to-profile alignment method that extends the accuracy of its intra-species MSAs into its inter-species MSAs. Through a cascading series of mapping methods, *Mirage2* is able to rapidly generate full-protein-length spliced mappings for nearly all of the human, mouse, and rat sequences in the *SwissProt* dataset. *Mirage2*'s high mapping success rate results in MSAs that display quality improvements over MSAs produced by existing alignment tools.

## Supporting information

**S1 File. A compressed directory containing scripts and instructions for replicating the results presented in this paper.**
(ZIP)

## Acknowledgments

George Lesica provided invaluable contributions towards improving *Mirage2*'s documentation, ease of installation, and overall accessibility. We are grateful for the use of the GSCC cluster at the University of Montana, without which, these analyses could not have been performed.

## Author Contributions

**Conceptualization:** Alexander J. Nord, Travis J. Wheeler.

**Funding acquisition:** Travis J. Wheeler.

**Investigation:** Alexander J. Nord.

**Methodology:** Alexander J. Nord, Travis J. Wheeler.

**Resources:** Travis J. Wheeler.

**Software:** Alexander J. Nord.

**Supervision:** Travis J. Wheeler.

**Validation:** Alexander J. Nord.

**Visualization:** Alexander J. Nord.

**Writing – original draft:** Alexander J. Nord.

**Writing – review & editing:** Travis J. Wheeler.

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
