## [Decision Letter · Decision Letter 0]

19 Jan 2023

PONE-D-22-34989Mirage2's high-quality spliced protein-to-genome mappings produce accurate multiple-sequence alignments of isoformsPLOS ONE

Dear Dr. Nord,

Thank you for submitting your manuscript to PLOS ONE. After careful consideration, we feel that it has merit but does not fully meet PLOS ONE’s publication criteria as it currently stands. Therefore, we invite you to submit a revised version of the manuscript that addresses the points raised during the review process.

We look forward to receiving your revised manuscript.

Kind regards,

Sheikh Arslan Sehgal, PhD

Academic Editor

PLOS ONE

Journal Requirements:

"We also acknowledge funding from the National Institute of General Medical Sciences (NIGMS), National Institutes of Health (NIH), grants GM123487 and GM132600, and from the National Human Genome Research Institute (NHGRI, NIH), grant HG012283."

"A.J.N. and T.J.W acknowledge funding from the National Institute of General Medical Sciences (NIGMS, https://www.nigms.nih.gov/), National Institutes of Health (NIH), grants GM123487 and GM132600, and from the National Human Genome Research Institute (NHGRI, NIH, https://www.genome.gov/), grant HG012283. The funders had no role in study design, data collection and analysis, decision to publish, or preparation of the manuscript."

**Additional Editor Comments:**

Mirage has been updated as Mirage2 to build MSA of isoforms by high quality spliced protein to genome mappings. The following points will improve the quality of the manuscript.

Authors did not mention the advantages of the new version and how new version is more effective in comparison to old version.

Figures are of poor quality.

Authors should cite the updated literature.

Discussion section is poorly drafted.

How the said software is better than already available software specifically BLAT.

Reviewers' comments:

Reviewer's Responses to Questions

**Comments to the Author**

1. Is the manuscript technically sound, and do the data support the conclusions?

Reviewer #1: Partly

Reviewer #2: Yes

2. Has the statistical analysis been performed appropriately and rigorously? 

Reviewer #1: Yes

Reviewer #2: Yes

3. Have the authors made all data underlying the findings in their manuscript fully available?

Reviewer #1: Yes

Reviewer #2: Yes

4. Is the manuscript presented in an intelligible fashion and written in standard English?

Reviewer #1: Yes

Reviewer #2: Yes

5. Review Comments to the Author

Reviewer #1: This manuscript provides a new program Mirage2, an updated version of Mirage, to build Multiple Sequence Alignments (MSAs) with protein-to-genome mapping. Mirage2 can map proteins to the encoding exons and the protein-genome can thus provide additional information when building MSAs compared to other general methods. Experimental results show that Mirage2 have advantage in both mapping and MSA construction.

However, the paper comes with a number of issues which the authors need to address.

1. The biggest concern is that this manuscript is very similar to Mirage (Ref 15). The differences between Mirage2 and Mirage are not clearly described. In abstract, the authors claims that Mirage2 provides substantially improved translated mapping, however, I did not see the difference in methodology between Mirage2 and Mirage in mapping. What will be the performance for Mirage for mapping and MSA construction? Will Mirage and Mirage2 have the same performance or not? If not, what are the differences? It is suggested to make a systematic comparison between Mirage2 and Mirage and highlight the updates.

2. How —blat-off will affect the MSA quality?

3. Another important application of MSA is the protein structure prediction. I wonder the mapping-enhanced MSA could improve the structure prediction quality by, for example AlphaFold2, compared with other controlled MSA build methods. This could largely improve the manuscript if it can help.

4. It is not necessary to show Fig 2 in the manuscript. A detailed example in the software package would be better for demonstration.

Reviewer #2: Nord and Wheeler updated the new version of Mirage i.e. Mirage2 to build multiple sequence alignments of isoforms by high qiality spliced protein to genome mappings. however the manuscript need revison.

The manuscript mainly lacks the diferences and advantages of Mirage 2 as compared to previous Mirage. The comprehensive comparsion should be included in the revised version to highlights its updates/improvements.

The figure 2 looks incomplete or more description need for this figure.

The references in text are not properly cited.

The manuscript is very limited particularly in introduction and discussion sections. incorporate more data of already developed tools and its features.

The preference of Blat needs to be discussed more.

6. PLOS authors have the option to publish the peer review history of their article (what does this mean?). If published, this will include your full peer review and any attached files.

Reviewer #1: No

Reviewer #2: No

---

## [Author Response · Author response to Decision Letter 0]

3 Apr 2023

We thank the reviewers for their thoughtful feedback on our manuscript, “Mirage2’s high-quality spliced protein-to-genome mappings produce accurate multiple-sequence alignments of isoforms”. Following the recommendations provided by the reviewers and editor, we have made the following improvements to the manuscript (highlighted in red within the submitted revision):

1. We have expanded the description of the relevant background and state of the field in the Introduction section.

2. Our description of the function of BLAT, both generally and within the context of our decision to use it within Mirage2, is much more robust in the Methods section.

3. In addition to the benefits described in the Discussion section, we have added a quantitative comparison of the performance benefits of Mirage2 over the original Mirage implementation in the Results section, emphasizing that Mirage2 halves the number of unmapped sequences from the original tool and that we have significantly improved the number of sequences mapped using our preferred method (FastMap).

4. We have added information about the impact of deactivating BLAT on MSA quality to the Results section.

5. Minor updates have been made to several of the figures.

6. We have improved Figure 2 to indicate that it is complete as-is. Because of the importance of this novel file format to the use of Mirage2, we believe that it is valuable for Figure 2 to remain in the manuscript.

7. Funding information has been removed from the Acknowledgements section. We would like to ensure that the submitted Funding Statement now reads: 

"A.J.N. and T.J.W acknowledge funding from the National Institute of General Medical Sciences (NIGMS, https://www.nigms.nih.gov/), National Institutes of Health (NIH), grants GM123487 and GM132600, and from the National Human Genome Research Institute (NHGRI, NIH, https://www.genome.gov/), grant HG012283. The funders had no role in study design, data collection and analysis, decision to publish, or preparation of the manuscript."

8. We have adjusted the bibliography to meet Vancouver-style formatting guidelines.

9. We have made a minimal underlying dataset available at the following stable URL:

https://osf.io/7pgav/ (DOI: 10.17605/OSF.IO/7PGAV)

We hope that these are all welcome changes and that the improved manuscript satisfies the quality and formatting guidelines for publishing with PLOSOne.

---

## [Editor Report · Decision Letter 1]

18 Apr 2023

Mirage2's high-quality spliced protein-to-genome mappings produce accurate multiple-sequence alignments of isoforms

PONE-D-22-34989R1

Dear Dr. Nord,

We’re pleased to inform you that your manuscript has been judged scientifically suitable for publication and will be formally accepted for publication once it meets all outstanding technical requirements.

Kind regards,

Sheikh Arslan Sehgal, PhD

Academic Editor

PLOS ONE
---

## [Editor Report · Acceptance letter]

27 Apr 2023

PONE-D-22-34989R1 

*Mirage2*’s high-quality spliced protein-to-genome mappings produce accurate multiple-sequence alignments of isoforms 

Dear Dr. Nord:

I'm pleased to inform you that your manuscript has been deemed suitable for publication in PLOS ONE. Congratulations! Your manuscript is now with our production department. 

Kind regards, 

on behalf of

Dr Sheikh Arslan Sehgal 

Academic Editor

PLOS ONE